# Recent Advances in the Development of Anti-FLT3 CAR T-Cell Therapies for Treatment of AML

**DOI:** 10.3390/biomedicines10102441

**Published:** 2022-09-30

**Authors:** Maya Graham Pedersen, Bjarne Kuno Møller, Rasmus O. Bak

**Affiliations:** 1Department of Clinical Immunology, Aarhus University Hospital, 8200 Aarhus, Denmark; 2Department of Biomedicine, Aarhus University, 8000 Aarhus, Denmark

**Keywords:** FLT3, AML, leukemia, FMS-like tyrosine kinase-3, CAR T, CAR, chimeric antigen receptor, FLT3L

## Abstract

Following the success of the anti-CD19 chimeric antigen receptor (CAR) T-cell therapies against B-cell malignancies, the CAR T-cell approach is being developed towards other malignancies like acute myeloid leukemia (AML). Treatment options for relapsed AML patients are limited, and the upregulation of the FMS-like tyrosine kinase 3 (FLT3) in malignant T-cells is currently not only being investigated as a prognostic factor, but also as a target for new treatment options. In this review, we provide an overview and discuss different approaches of current anti-FLT3 CAR T-cells under development. In general, these therapies are effective both in vitro and in vivo, however the safety profile still needs to be further investigated. The first clinical trials have been initiated, and the community now awaits clinical evaluation of the approach of targeting FLT3 with CAR T-cells.

## 1. Introduction

The approval of the first chimeric antigen receptor T (CAR T) cell therapy revolutionized cancer treatment when it proved both possible and safe to use CAR T-cell therapy as a treatment option for cancer patients. Since the first CAR T-cells were described nearly 20 years ago [1], the research field has grown significantly and was accelerated by the FDA approval in 2017 of the anti-CD19 CAR T-cell therapy for acute lymphoblastic leukemia (ALL) and non-Hodgkin lymphoma (NHL) [2,3,4]. In June 2022, six different CAR T-cell therapies were FDA approved [5]. Four of these therapies target CD19 and two target the B-cell maturation antigen (BCMA) to treat multiple myeloma [6]. Many other CAR T-cell therapies are in development that target other antigens besides the well-known CD19. Currently, CAR T-cell therapies to treat malignancies like myeloid leukemias or myelodysplasia (targeting antigens like CD123 and CD33) are underway, while other treatment modalities focus on a bispecific CAR design targeting both CD19 and CD20 or CD22 [7]. This review focuses on current CAR T-cell research and clinical trials specifically targeting the FLT3 antigen to treat acute myeloid leukemia (AML).

## 2. Background on AML

AML is a hematological malignancy affecting the differentiation of myeloid cells causing a buildup of immature cells. The interrupted blood cell maturation causes symptoms of thrombocytopenia, neutropenia, and anemia [8,9]. Current treatment options consist of induction chemotherapy, typically 7 days of cytarabine and 3 days of anthracycline [10]. The initial response (complete remission) is 60–80% for patients below 60 years and 40–60% above 60 years of age [10,11,12]. Although there is a relatively high complete remission rate, the 5-year overall survival is approximately 30% because one third of AML patients relapse after their first line treatment of chemotherapy [13,14,15]. For relapsed patients, the only curative option is hematopoietic stem cell transplantation (HSCT). Older patients (>60 years) with decreased performance status are not always eligible for HSCT and therefore have limited treatment options [12,14]. Of the patients above >60 years who are eligible for HSCT, the 5-year overall survival after HSCT is 35% with a non-relapse mortality of 18% [16]. Among general adult AML patients (>18 years) who are minimal residual disease (MRD) positive when transplanted, the relapse rate is high with two thirds relapsing in the first three years after transplantation with an overall 3-year survival of around 20% [17]. In complete remission of MDR negative patients, the prognosis is better with overall 3-year survival of 73% and relapse rate of 22% [17]. The non-relapse mortality associated with HSCT is caused by complications like organ toxicity, infections, and graft versus host disease (GvHD) [18]. Hence, new treatment options for AML are desperately needed. CAR T-cells directed towards the FMS-like tyrosine kinase-3 (FLT3) antigen present on AML cells could be a viable treatment option. CAR T-cell preclinical studies targeting FLT3 for treatment of AML are listed in Table 1, and ongoing clinical trials are listed in Table 2.

## 3. Configuration of Chimeric Antigen Receptors

The chimeric antigen receptor (CAR) is an artificial antigen receptor specifically redirecting the T-cells to the chosen target. A CAR is composed of three domains: an extracellular, transmembrane, and intracellular signaling domain (SD). The extracellular domain is typically derived from a known monoclonal antibody or developed specifically for the target antigen. It consists of a single-chain variable fragment (scFv) composed of a variable light (VL) and variable heavy chain (VH). It is possible to even make the CAR bispecific by adding two different scFvs so that the CAR T-cells recognize two different epitopes either on the same or on different receptors. Through a hinge, the CAR is connected to the transmembrane (TM) domain, which links the extracellular domain with the intracellular domain. The addition of co-stimulatory domains in the intracellular SD typically determines the CAR generation. First generation CARs only consist of the CD3ζ SD, which is derived from the endogenous T-cell receptor (TCR). CD3ζ was not sufficient to fully activate CAR T-cells [19]. In second generation CARs, co-stimulatory domains like CD28 or 4-1BB were added to the intracellular SD. Other co-stimulating domains have also been used, including OX40, CD27, and inducible T-cell co-stimulator (ICOS) [20]. In third generation CARs, both CD28 and 4-1BB domains are added as intracellular SDs to enhance persistence, proliferation, and efficacy. It has been argued that third generation CARs do not enhance efficacy compared to second generation CARs, but Zhang et al. argue that sufficient direct comparisons of the second and third generation CARs are still needed to conclude this, and that it might be dependent on the specific target antigen [21]. The different anti-FLT3 CAR-constructs which have been preclinically tested are schematically represented in Figure 1.

The choice of costimulatory domains influences T-cell phenotype, persistence, cytokine production, and proliferation. The most frequent co-stimulatory domains used are the 4-1BB and CD28, and some studies report one to be more advantageous than the other, but presently without agreement [22,23]. Long et al. compared the two costimulatory domains CD28 and 4-1BB and found that the anti-CD19 CAR harboring 4-1BB showed lower levels of exhaustion compared to CD28, due to tonic CD3ζ signaling caused by antigen-independent receptor activation [22]. Drent et al. also examined the difference between 4-1BB and CD28, but for treatment of multiple myeloma [24]. When using a 4-1BB co-stimulatory domain, they observed reduced differentiation, less exhaustion, and improved proliferative capacity compared to CD28. However, the observed effects were dependent on the specific scFvs and the affinity for the target antigen [25]. Studies from Salter et al. support these findings. In vitro CD28 CARs were activated faster and with larger magnitude, whereas 4-1BB CAR T-cells preferentially expressed T-cell memory genes and generally displayed more sustained cytotoxic activity [26]. Despite different comparisons between the two co-stimulatory domains, it does not currently seem possible to generally determine one to be superior to the other [27,28]. Clinically, anti-CD19 CAR T-cells with CD28 or 4-1BB as co-stimulatory domains have different peak time of cytokine production, and it is recognized that they show differential engagement of intracellular signaling pathways, but no significant differences are observed with regard to CAR expression, cytotoxicity, cytokine production (IL-6, IL-10, IFN-γ, and IL-2 secretion), expression of exhaustion markers (Tim-3 and LAG-3), and T-cell phenotypes [28,29]. In conclusion, it is evident that a CAR needs to be at least a second-generation CAR, but no superior consensus CAR configuration has been devised and the CAR should therefore be specifically designed and optimized for the target antigen.

**Figure 1 biomedicines-10-02441-f001:**
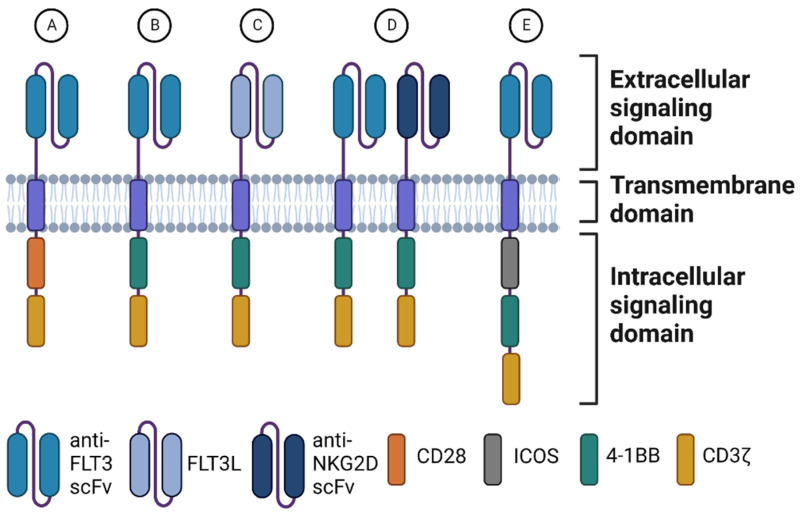
An overview of different anti-FLT3 CAR T-cells developed to date. (**A**) A second-generation CAR with CD28 as co-stimulatory domain [30,31,32]. (**B**) A second-generation CAR with 4-1BB as co-stimulatory domain [30,33]. (**C**) A second-generation CAR with 4-1BB as co-stimulatory domain and the FLT3 ligand (FLT3L) instead of an anti-FLT3 scFv [34]. (**D**) Two CARs with scFvs targeting FLT3 and NKG2DS, respectively. The CARs were encoded on a bicistronic lentiviral vector using the same promoter and separated by a self-cleaving 2A peptide to ensure equal expression on the cell surface [35]. (**E**) A third generation anti-FLT3 CAR with two co-stimulatory domains, 4-1BB and inducible T-cell co-stimulator (ICOS) [36].

## 4. The Choice of FLT3 as Target for CAR T-Cell Therapy

For the development of any CAR T-cell therapy, an appropriate scFv must be chosen that binds to a surface antigen on the target cells. Ideally, the targeted antigen should be expressed at high levels to improve efficacy of the CAR T-cells while being minimally expressed on non-cancer cells to limit off-tissue toxicity.

Different proteomic and transcriptomic strategies have been taken to identify suitable antigens of proteins with high expression on tumor cells with minimal expression on normal tissue. Perna et al. used large surface genome datasets from malignant (AML) and normal tissues and found CD123, CLEC12A, and CD33 to be highly expressed on AML blasts at >75% expression but also with high expression on normal hematopoietic stem cells with the possibility of introducing many side effects in patients [37]. Despite this risk, several CAR T-cells for AML are in development targeting CD33, CD123, and CLEC12A, and patients are currently being recruited for clinical trials [38]. FLT3 is also expressed on other hematologic malignancies, particularly B-lineage acute lymphocytic leukemia (ALL), and it has also been reported to be amplified on some solid tumors like breast cancer, colorectal cancer, and gastric cancer [39]. However, the importance of FLT3 in solid tumors remains less studied.

The FMS-like tyrosine kinase-3 (FLT3) receptor is normally expressed on a fraction of CD34+ hematopoietic stem and progenitor cells (HSPCs) and plays important roles in regulating hematopoiesis through control of survival, proliferation, and differentiation [40,41]. Both membrane-bound and cytoplasmic FLT3 protein have been identified with glycosylation governing membrane localization [42]. FLT3 has been found to be expressed on 40–50% of the examined AML patients based on two different studies totaling 1252 patients [43,44]. In contrast to CD123, CLEC12A, and CD33, FLT3 surface expression is only evident in HSPCs, thereby limiting some potential toxicities, but retaining a risk of depleting the HSPC compartment [40]. Furthermore, elimination of FLT3-positive cells may interfere with important signaling through the FLT3/FLT3L axis, which is important especially for B-cell development [45]. In FLT3-positive AML patients, mutations have been observed in the FLT3 gene on chromosome 13. The most prevalent of the FLT3 mutations is the internal tandem duplication (ITD) which is found in 15–30% of the AML patients [44,46,47,48]. The FLT3-ITD is an in-frame duplication in the juxtamembrane domain of the intracellular region of the FLT3 receptor (Figure 2). While FLT3 receptor activation normally occurs through binding of the FLT3 ligand, the ITD mutation renders the receptor constitutively active thereby driving leukemogenesis by promoting survival and proliferation [49,50]. The second most prevalent FLT3 mutation is in the tyrosine kinase domain (TKD), which 5–10% of AML patients harbor [48,51]. The FLT3-ITD mutations are associated with reduced overall survival and are predictors of poor prognosis [43,46,51,52]. Thiede et al. found that the risk of relapse was 1.6 with a median ratio (ITD/WT genotype) above 0.78 and poor overall survival, which is consistent with Whitman et al. that found an overall survival rate of 74% for the FLT3 WT/WT genotype compared to 13% (*p* = 0.008) for the FLT ITD/-genotype [48,53]. The prognostic significance of the FLT3-TKD is still unclear because no statistically significant worse overall survival has been shown compared with the FLT3-ITD [48,54].

The poor prognosis of AML patients harboring FLT3 mutations led to the development and FDA approval of several small molecule inhibition therapies validating FLT3 as a target in FLT3-positive AML patients. The first generation FLT3 inhibitors (sorafenib, midostanin, lestaurtinib, sunitinib, and tandutinib) were significantly less selective than the subsequent second-generation inhibitors (gilteriteinib, quizartinib, and crenolanib). The main problem associated with these small molecule inhibition therapies has been development of resistance to the treatment by the AML cells. CAR T-cell therapy might be an advantage because of its ability to permanently kill AML cells. Furthermore, an optimal anti-FLT3 CAR T-cell therapy should target the FLT3 receptor independently of the mutational status and should avoid killing normal HSPCs, if possible. Figure 2 provides an overview of where the different anti-FLT3 CARs bind in relation to the most dominant mutations.

In summary, no antigen is exclusively expressed in AML cells without expression in normal cells. In addition, FLT3 expression in AML cells is associated with poor prognosis, and higher FLT3 density associated with relapse. These observations have been used to argue that FLT3 represents an ideal target for CAR T-cell therapy, either as a bridging therapy before hematopoietic transplantation or ideally as a curative intervention.

## 5. In Vitro Assessment of Efficacy and Safety of Anti-FLT3 CAR T-Cell Therapies

Several different approaches have been used to construct anti-FLT3 CAR T-cells. Some studies use previously developed anti-FLT3 antibodies where the heavy and light chains of the antibody are used as an scFv for the extracellular part of the CAR. The intercellular part of the CAR construct, such as the choice of co-stimulatory domain, can be made to mimic the anti-CD19 CAR T-cells, which are known to be effective.

Table 1 presents seven anti-FLT3 CAR T-cell therapies currently described in the literature. While most studies used an scFv derived from known antibodies, Sommer et al. developed and screened their own scFvs from a phage-display library [33]. In contrast to using scFvs, Wang et al. and Maiorova et al. targeted FLT3 using its natural ligand, FLT3 ligand (FLT3L) [34,36]. This has the advantage of potentially eliminating problems with immunogenicity, and the natural ligand might display higher specificity than scFvs. The seven studies also use different configurations of anti-FLT3 CARs, mono- and bispecific CAR T-cells, and second and third generation CARs with different co-stimulatory domains, ICOS, 4-1BB, and CD28. All CAR T-cells were produced by lentiviral transduction with variable CAR expression ranging from 24 to 90%. Jetani et al. constructed two different anti-FLT3 CAR T-cells using either 4-1BB or CD28 as costimulatory domains [30]. They found that the anti-FLT3 CAR T with the CD28 costimulatory domain produced higher levels of IL-2 and displayed superior proliferation capacity. No further comparison of the two different CAR T-cells was conducted but based on their findings they continued with CD28 as co-stimulatory domain. All second generation anti-FLT3 CAR T-cells with either CD28 or 4-1BB showed cytotoxicity against different FLT3-positive cell lines; Kasumi, OCI-AML3, MOLM-13, THP-1, EOL-1 or MV4-11. Maiorova et al. chose to use ICOS and 4-1BB to generate a third generation CAR [36]. The authors used mKate2-expressing (red fluorophore) THP-1 cells and measured decreased red object count as a representation of THP-1 killing. The red object count of THP-1 was reduced to around zero at both E:T = 1:1 and E:T = 5:1 after 70 h in contrast to red object count of FLT3-negative U937 cells, and the authors therefore concluded that the third generation anti-FLT3 CAR T-cells were effective with no exact percentage of lysis presented.

As previously stated, the antigen recognition region of the anti-FLT3 CARs used in the studies are mainly from previously developed monoclonal antibodies. Sommer et al. took a different approach by producing several different anti-FLT3 CAR T-cells [33]. They first used surface plasmon resonance to identify appropriate scFvs that bind FLT3 protein efficiently. Thereafter, they used the nine most promising scFvs to develop anti-FLT3 CAR T-cells and test their functionality. They cocultured FLT3-positive cells with CAR T-cells at an effector to target ratio of 1:1 and added fresh target cells continuously to examine tonic signaling, expansion, and cytotoxicity of the CAR T-cells during a 20-day period. They measured the viability of the target cells at different time points and CAR T-cell expansion at the end of the 20 days. Tonic signaling, i.e., CAR activation without antigen recognition, was measured by percentage of CAR T-cells showing activity of the co-stimulatory domain 4-1BB six days after transduction. Among the nine different CAR T-cells, the highest frequency of CAR T-cells showing 4-1BB activity due to tonic signaling was around 15%. The different CAR T-cell populations were further examined for their T memory stem cell phenotype, which was compared to untransduced T-cells which have around 50% T memory stem cells for both CD4+ and CD8+ cells [33]. The best anti-FLT3 CAR T that were chosen for further evaluation showed low tonic signaling (3–6% of cells with 4-1BB activity), less differentiated with the largest proportion of the T memory stem cell phenotype (~25% for both CD4+ and CD8+), high fold-expansion (11 to 16-fold), and cytotoxicity against different target cells (80–100% lysis against EOl-1, MOLM-13, and MV4-11).

None of the other studies provided this detail of anti-FLT3 CAR T-cell characterization. Wang et al. also developed a CAR with 4-1BB as co-stimulatory domain and using the natural FLT3 ligand (FLT3L) to target FLT3 [34]. The CAR T-cell phenotype was measured seven days after transfection and compared with unmodified T-cells. They found a significantly increased proportion of central memory T-cell phenotype compared with the T-cells carrying an empty vector. In addition to this, they identified no significant differences in proportions of effector memory, terminally differentiated effector memory, or naïve T-cell phenotypes between the FLT3L CAR T-cells and unmodified T-cells. They did not examine T-cell memory stem cell phenotype, and no other authors examined anti-FLT3 CAR T-cell phenotype or exhaustion [34].

The cytotoxicity of anti-FLT3 CAR T-cells in vitro against FLT3-positive cell lines varied between the studies from specific lysis of 10–100% at effector to target (E:T) ratios from 10:1 down to 3:1 (Table 1). Sommer et al. already observed 100% specific lysis at E:T = 3:1 and reported the lowest E:T ratio of 1:6 where they observed specific lysis of EOL-1, MOLM-13, and MV4-11 of around 50% [33]. Cytotoxicity levels observed by Wang et al. are not displayed in the table because they measured the percentage of surviving target cells. At the E:T = 1:4 they found 0% live MV4-11 cells, 0% live MOLM-13 cells, 25% live REH cells, and 60% live THP-1 cells [34]. Killing capacity did not correlate with FLT3 surface levels and THP-1 cells were effectively killed by anti-FLT3 CAR T-cells in another study [30].

The THP-1 monocytic cell line carries non-mutated FLT3 and is therefore not very dependent on FLT3 receptor signaling. It will respond to stimulation by the FLT3L as observed by Maiorova et al. but is not dependent on it [36,61]. Wang et al. found that the effect of the FLT3L CAR T-cells was not dependent on FLT3 expression, but rather on whether the FLT3-ITD mutation was present [34]. The cytotoxic effect of the FLT3L CAR T-cells was greater towards FLT-ITD (MV4-11, MOLM-13) compared to WT FLT3 (THP-1, REH) [34]. They measured FLT3 signaling upon co-culture with FLT3L CAR T-cells and found increased ERK phosphorylation (pERK) in FLT3-WT indicating downstream activation of the FLT3-WT AML cells. It is unclear if this activation causes the decrease in cytotoxicity against FLT3-WT. Jetani et al. found high expression of FLT3 protein to influence the effectiveness of the anti-FLT3 CAR T-cells [30]. They observed an increased cytotoxicity against THP-1 and MOLM-13 cells which have higher FLT3 levels compared to MV4-11. However, they found no difference in the cytotoxicity against FLT3-WT and FLT3-ITD. Sommer et al. examined patient blasts (*n* = 3) and measured frequencies of FLT3-positive cells of 60–80% with no difference between FLT3-WT and FLT3 mutation (FLT3-MUT). They showed an elimination of 80% of the AML patient cells in co-cultures with anti-FLT3 CAR T-cells [33].

FLT3 is expressed on normal HSPCs, and an important safety parameter for the produced anti-FLT3 CAR T-cell therapies is therefore in directly assessing lysis of normal HSPCs or conducting a colony forming unit (CFU) assay of the HSPCs to quantify loss of colony formation as a sign of cytotoxicity against progenitor cells. Wang et al. found that 80–90% of isolated CD34+ HSPCs from cord blood expressed FLT3 and found no significant difference in FLT3 expression levels between HSPCs, AML patient cells, and an AML cell line [34]. Wang et al. performed a CFU assay and found that the FLT3L CAR T-cells did not cause loss of colony formation [34]. Similar results were found by Chen et al. who did not observe cytotoxicity of the anti-FLT3 CAR T towards cord blood CD34+ HSPCs using a 51Cr release assay and analysis of IFN-γ secretion [31], whereas both were observed when using the FLT3-positive MOLM-13 cell line as a positive control. These observations by Wang et al. [34] and Chen et al. [31] contrast with the rest of the articles that did detect HSPC killing (Table 1). Jetani et al. found that their FLT3 CAR T-cells eliminated 50% of HSPCs in the first four hours of co-culture and 80% in 24 h (E:T = 5:1) [30]. These anti-FLT3 CAR T-cells were furthermore compared to anti-CD123 CAR T-cells that eliminated 95% of HSPCs in 24 h. Li et al. [35] found 20% lysis of HSPCs, and Sommer et al. [33] found a significant reduction without specifying the exact percentage.

To summarize, it is possible to create anti-FLT3 CAR T-cells that effectively kill FLT3 positive cells in vitro. No evidence points towards specificity towards the different FLT3 mutations and none of the developed therapies have conclusively been proven to be superior to others.

## 6. In Vivo Assessment of Efficacy and Safety of Anti-FLT3 CAR T-Cell Therapy

Immunodeficient (NOD/SCID) mice have for the most part been used to study efficacy and safety of anti-FLT3 CAR T-cell therapies. The efficacy is determined by survival of the animals (humane endpoint) and tumor burden by bioluminescence using an in vivo imaging system. Chen et al. used MV4-11 target cells to define the effective dosage (2 × 10^5^ or 2 × 10^6^) of anti-FLT3 CAR T-cells [31]. Both low and high doses showed a decline in tumor burden, but the latter was found to be most effective. The efficacy was thereafter demonstrated using MOLM-13 target cells and AML patient cells (containing around 90% positive FLT3 AML blasts). In mice engrafted with MOLM-13 cells or patient AML blasts, the survival was 100% at day 80 (control mice were euthanized at day 25 because of high tumor burden) and 120 days respectively (control mice were euthanized at day 90 because of high tumor burden) (*p* < 0.001). Anti-FLT3 CAR T-cells could be measured at 49 days post-infusion, but not at day 84. Jetani et al. (monospecific anti-FLT3 CAR T-cells) and Li et al. (bispecific FLT3/NKG2DS-CAR T-cells) found similar results, with a prolonged survival of MOLM-13 injected mice compared with the control group (*p* < 0.05) [30,35]. Furthermore, Jetani et al. evaluated CAR T-cell efficacy with the addition of FLT3 small molecule inhibition therapy [30]. When the anti-FLT3 CAR T-cell therapy was administered as monotherapy, an overall response rate of 75% was observed with no cancer cells detectable in mice sacrificed at day 21 to 28. The control group did not survive past day 15 and the bone marrow had a frequency of 50–70% MOLM-13 cells. There was also an improved survival with anti-FLT3 CAR T-cells administered in combination with crenolanib compared to anti-FLT3 CAR T-cells as monotherapy (*p* < 0.005). Li et al. who used bispecific FLT3/NKG2D-CAR T-cells found a significant survival improvement with CAR T-cell therapy, with a median survival of 24 days compared to the control group of 15 days (*p* < 0.05) [35]. When combined with second generation gilteritinib, they also detected a significant benefit in median survival of 35 days compared to monotherapy with a median of 24 days (*p* < 0.05). Wang et al. used MV4-11 target cells and the FLT3L CAR T-cells prolonged the survival of the mice to 126 days compared with the control group of 86 days (*p* = 0.0039, *n* = 7 pr group) [34]. Sommer et al. used NSG mice engrafted with luciferase-labeled EOL-1 cells to evaluate the anti-FLT3 CAR T-cell therapy [33]. Four different anti-FLT3 CAR T-cells had been chosen from the in vitro studies. All four were evaluated in vivo, but only two showed antitumor activity (P3A1 and P3E10). The mice were tumor free after 35 days after a single dose of anti-FLT3 CAR T-cells. When the anti-FLT3 CAR T-cells were produced from two different donors and assessed in vivo, P3E10 was superior. The mice showed significantly prolonged median survival of >45 days compared to the control group < 25 days and decreased tumor burden (*p* < 0.001).

For the in vivo safety assessment, Chen et al. injected anti-FLT3 CAR T-cells into NSG mice simultaneously with human HSPCs and observed no detectable reduction in the ability of the HSPCs to engraft and humanize the mice (measured at 1 and 3 months post transplantation) compared with the control group [31]. In contrast, Jetani et al. engrafted their mice with human HSPCs for eight weeks prior to anti-FLT3 CAR T-cell injection and found depletion of normal HSPCs [30]. These in vivo results agreed with in vitro results of the same study.

Sommer et al. also demonstrated in vitro toxicity towards HSPCs (both human and mouse) and sought to improve safety of their anti-FLT3 CAR T-cells by including a safety off switch (R2), a CD20 epitope targetable by rituximab [33]. They positioned this off switch into the hinge region of the anti-FLT3 CAR T and verified that it did not alter the efficacy before administering the cells in vivo. They demonstrated that the anti-FLT3 CAR-R2 T-cells could eradicate the cancer cells and that the CAR T-cells were partially depleted when rituximab was administered. The partial depletion of CAR T-cells was sufficient to reconstitute mouse HSPCs. No relapse of the cancer was observed 15 days after depletion of the CAR T-cell therapy. Furthermore, using gene editing based on TALE nucleases they knocked out TRAC (the constant region of the T-cell receptor) to produce TCR-negative allogeneic anti-FLT3 CAR T-cells. They found similar antitumor activity in vivo of CAR T-cells with TRAC knockout compared to non-edited CAR T-cells.

Karbowski et al. present the only study using cynomolgus monkeys to demonstrate tolerability, safety, and dose-dependent efficacy vs. toxicity [32]. Karbowski et al. sought to mimic the human environment as closely as possible to improve prediction of clinical performance and safety of their anti-FLT3 CAR T-cell therapy [32]. They specifically examined the monkeys for FLT3 expression in tissues that were previously documented to express the FLT3 protein. They concluded that FLT3 protein, which has been previously documented to be expressed in the kidney, pancreas, prostate, cortical neurons, and hepatocytes, is cytoplasmic in these tissues and therefore not accessible to the anti-FLT3 CAR T-cells. However, as in the human setting, they also confirmed that CD34+ HSPCs expressed FLT3 on the surface and might therefore be subject to the cytotoxic effects of anti-FLT3 binding molecules as demonstrated by other studies except for Chen et al. and Wang et al. [31,34]. Monkey T-cells carrying the anti-human FLT3 CAR were shown to exhibit cytotoxicity towards cells carrying non-human primate FLT3. In vivo, administration of these CAR T-cells did not lead to expansion of the CAR T-cells and the CAR T-cells were not detectable at the end of the study, thereby indicating no or minimal engagement of endogenous FLT3-positive cells. Clinical adverse effects in the monkeys were short-term and limited to an increase in body temperature and increased C-reactive protein (CRP), but both parameters returned to baseline at day 4. Based on the results from this safety assessment of the anti-FLT3 CAR T-cells, the investigators have proceeded to initiate a clinical trial (NCT03904069). Two additional clinical trials using anti-FLT3 CAR T-cells are also reported to be recruiting (Table 2) [31,34].

In conclusion, anti-FLT3 CAR T-cell therapies have been demonstrated to prolong the survival of treated mice compared with the control group. In terms of safety, on-target, off-tumor reactivity is a concern also for anti-FLT3 CAR T-cells. As signaling through the FLT3/FLT3L axis is imperative for functional lymphopoiesis, clinical application might rely on replenishment of these lineages following elimination of anti-FLT3 CAR T-cells for example using a kill switch, or replenishment from an allogeneic transplantation if using the CAR T-cells as a bridging therapy [45].

**Table 1 biomedicines-10-02441-t001:** Overview of anti-FLT3 CAR T-cells derived from text or figures of the referenced articles.

Reference	Antigen Recognition Domain	CAR Generation	Co-Stimulatory Domain (s)	Proportion of T-Cells with CAR Expression	Cytotoxicity on Cell Line In Vitro E/T	Cytotoxicity on Primary AML Cells In Vitro	Difference Observed in Cytotoxicity between FLT3 Genetic Variants	Lysis of Normal HSPCs	In Vivo
Chen et al. [31]	Anti-FLT3 monoclonal antibody clone 4G8 [59]	2nd	CD28	80–90%	Kasumi: ~7% OCI-AML3: ~12% (E:T = 10:1)	40–45% (*n* = 4, FLT3+, ~90%) 28–40% (*n* = 4, FLT-ITD)	No difference observed	No lysis observed	NSG
Jetani et al. [30]	Anti-FLT3 monoclonal antibody clone 4G8 [58]	2nd	CD28 or 4-1BB	>90% *	MOLM-13: ~79% THP-1: ~85% MV4-11: ~60% (E:T = 10:1)	>80% (*n* = 3, 2/3 FLT3− ITD+)	Higher MFI -> more cytotoxicity	50–80% (E:T ratio = 5:1)	NSG
Karbowski et al. [32]	Anti-FLT3 scFv from earlier research [62]	2nd	CD28	24–74%	N/A **	N/A	N/A	N/A	Cynomolgus monkeys
Li et al. [35]	Two CARs targeting FLT3 and NKG2D, respectively. FLT3 scFv derived from monoclonal anti-FLT3 antibody clone EB10 [60]	2nd	4-1BB	30%	MOLM-13: ~27% MV4-11: 27% (E:T = 10:1)	N/A	Significant > killing of FLT− MUT+ compared with FLT-MUT-	7% (E:T = 1:1) 20% (E:T = 10:1) 23% (E:T = 20:1)	NSG
Maiorova et al. [36]	Full-length human FLT3 ligand	3rd	ICOS and 4-1BB	49%	N/A **	N/A	N/A	N/A	N/A
Sommer et al. [33]	Various anti-FLT3 scFvs from a phage library were screened for binding probabilities.	2nd	4-1BB	30–60%	EOL-1: ~100% MOLM-13: ~100% MV4-11: ~100% (E:T = 3:1)	Around 80% lysis observed	N/A	Significant reduction in HSPCs, not specified in %. (E:T = 1:1)	NSG
Wang et al. [34]	The binding domain of human FLT3 ligand.	2nd	4-1BB	40–50%	Measured survival of target: E:T = 1:4 MV4-11: 0% MOLM-13: 0% REH: 25% THP-1: 60% (E:T = 10:1)	Live AML cells after anti-FLT3L CAR T: 5 FLT IDT = 5–30% live AML cells 5 FLT3 WT = 70–20% live AML cells	Higher cytotoxicity against FLT3−ITD	No lysis observed	NSG

* CAR T-cells were enriched to this purity using a truncated EGFR selection marker ** The cytotoxicity of effector to target ratio (E:T) of 1:10 was not performed.

**Table 2 biomedicines-10-02441-t002:** Anti-FLT3 CAR T-cell trials [63].

Disease	Drug	Phase	Status	Country/Sponsor	Clinical Trial Identification
FLT3-positive relapsed/refractory AML	Anti-FLT3 CAR T	I/II	Recruiting	China/The First Affiliated Hospital of Soochow University	NCT05023707
FLT3-positive relapsed/refractory AML	AMG 553 (anti-FLT3 CAR T)	I	Not yet recruiting	USA/Amgen	NCT03904069
Recurrent/refractory FLT3 positive AML	TAA05 (anti-FLT3 CAR T)	N/A	Recruiting	China/PersonGen BioTherapeutics (Suzhou) Co., Ltd.	NCT05017883

## 7. Discussion

Anti-FLT3 CAR T-cells have been demonstrated to be both effective in vitro and in vivo. The CAR design, which has been proven to be effective in CD19 CAR T-cell therapies, has also been applied in the design of anti-FLT3 CAR T-cells. Sommer et al. demonstrated that CAR expression affects the T-cell phenotype and Wang et al. had similar findings with a significant increase of central memory cells demonstrating a shift in T-cell phenotype [33,60]. It will therefore be important to determine if the CAR T-cells are constitutively activated and what this means for long-term persistence of these cells in a clinical setting. They found that the anti-FLT3 CAR T-cells, which had a high activation level, were also highly differentiated. Sommer et al. illustrated this by demonstrating that the proportion of CAR T memory stem cells were approximately halved compared to unmanipulated T-cells [33]. Likewise, they observed that the CAR T-cells which were highly active due to the co-stimulatory domain also were the most differentiated. Overall, CAR construct configuration has an impact on T-cell phenotype and generating less differentiated CAR T-cells retaining a significant proportion of stem cell memory or central memory cells are needed to provide long-term persistence. Effector CAR T-cells may be effective short-term, but if the tumor burden is too high or expanding too fast, the CAR T-cells might be exhausted before total eradication of tumor cells. One way to manipulate the phenotype is to manufacture the cells under conditions that promote a memory stem cell phenotype, e.g., by supplementing the culture medium with or replacing IL-2 with IL-15 [64].

Several anti-FLT3 CAR T-cells have been developed, and the in vitro efficacies reported vary between 7–100% in the different studies. Some studies (Jetani et al., Li et al., and Wang et al.) show that the difference in the cytotoxic effect is dependent on the mutational status of the FLT3 receptor [30,34,35]. FLT3 mutations in malignant cells are positioned in the intracellular domains of the kinase, whereby signaling is de-regulated contributing to the malignant transformation of the cells. The intracellular location of the mutations makes them inaccessible to CAR T-cells and hinders mutation-specific approaches. Hence, this is an interesting observation and further studies should be conducted to determine the exact influence of the mutations on anti-FLT3 CAR T-cell efficacies.

FLT3 is present on normal HSPCs and Chen et al. and Wang et al. all showed that their anti-FLT3 CAR T-cells did not kill HSPCs [31,34]. Chen et al. used a different approach to measure the cytotoxicity of FLT3 CAR T-cells on HSPCs by injecting human HSPCs at the same time as injection of anti-FLT3 CAR T-cells. This contrasts with other studies that first allowed the human HSPCs to engraft in the mice [31]. As noted by Sommer et al., pre-engraftment of the HSPCs might mean that the anti-FLT3 CAR T-cells have less access to the HSPCs and that this could explain why no cytotoxicity was observed on human HSPCs [33]. Further research needs to be conducted to fully elucidate the observed differences between CAR T-cell cytotoxicity on HPSCs and malignant cells, and if it is possible to develop a CAR T-cell therapy that only targets AML cells without a major impact on HSPCs. If this is not possible, Sommer et al. does provide a contingency strategy with the integration of an off switch that may allow shutting down CAR T-cell activity [33].

In vivo, the different studies demonstrated a prolonged survival of the mice injected with anti-FLT3 CAR T-cells compared to the control group. To what extent these results will translate into the clinical setting is still to be seen. Compared with the other CAR T-cell therapies targeting CD33, CD123, and CLEC12A, the anti-FLT3 CAR T-cells have the same safety issues regarding potential depletion of the HSPC compartment [65,66]. These antigens are expressed at higher levels on the surface of AML cells compared to FLT3 and may therefore represent more effective target antigens [66]. FLT3 is typically upregulated because of prior treatment with small molecule inhibition therapies or chemotherapy that increase the FLT3 ligand level [67,68,69]. The FLT3 positive patients who relapse from other treatments will likely be more effectively treated with anti-FLT3 CAR T-cell therapy than salvage chemotherapy. Regarding clinical translation from the mouse studies, we can only hypothesize about the effects in a human setting. The immunodeficient mice used in in vivo studies lack a complete human immune system, and this deficit may confound results [40,70,71]. Other effects such as cytokine release, CAR T-cell proliferation and persistence, microenvironment modulation, and tumor resistance are all important parameters in the assessment of the possible effectiveness of anti-FLT3 CAR T-cells. The effectiveness of such treatment may be profoundly influenced by the proliferation rate and immunoinhibitory mechanisms of cancer cells as well as the CAR T-cells themselves, due to T-cell donor differences as well as differences in the specific protocols used to prepare the CAR T-cells.

## 8. Conclusions and Future Directions

Development of anti-FLT-3 CAR T-cell therapies is progressing rapidly with several clinical trials under way. The outcome of these trials will provide further insight into efficacy and safety of this novel class of CAR T-cells. However, basic research in FLT3 targeting is still very limited, and further research should be conducted to elucidate the best CAR design as well as the effect of FLT3 mutations on treatment efficacy. The challenge with AML is that there are no good antigen targets that are exclusively expressed on AML cells. With more research it might be possible to develop anti-FLT3 CAR T-cells that are fine-tuned to effectively target AML while sparing normal HSPC. Currently developed anti-FLT3 CAR T-cell therapies can be used as a bridge therapy before hematopoietic stem cell transplantation of FLT3-positive patients. Anti-FLT3 CAR T-cell therapy can be genetically engineered to harbor an off switch to deplete the therapy before transplantation or after the AML has been eliminated and normal immune reconstitution from FLT3-proficient HSPCs can occur. Novel genetic engineering technologies may further enhance CAR T-cell efficacy and enable allogeneic, off-the-shelf CAR T-cells [72]. Implementation of more advanced synthetic biology promises to enable more sophisticated programing of multi-antigen recognition and exclusion to further enhance safety issues and broaden CAR T-cells to tissues with more complex antigen composition [73].

## Figures and Tables

**Figure 2 biomedicines-10-02441-f002:**
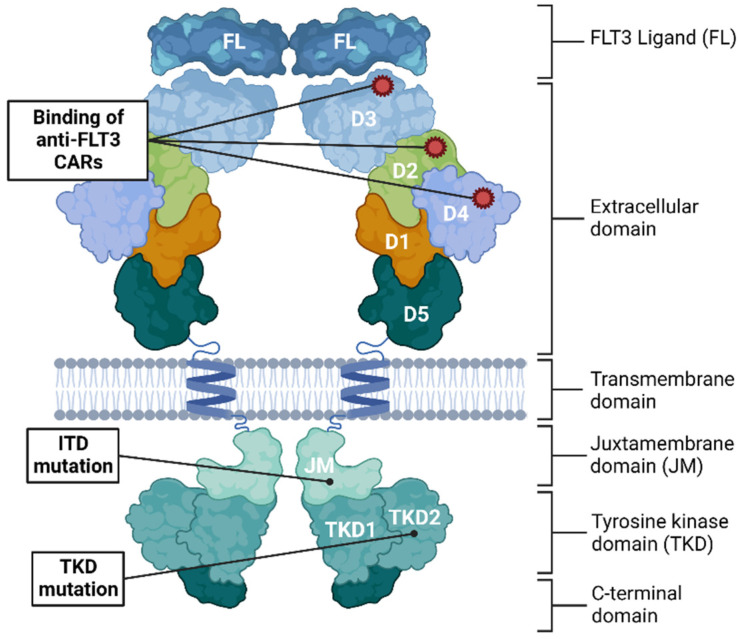
The structure of the FLT3 receptor and the FLT3 ligand (FL). FLT3 consists of an extracellular segment organized in five Ig-like domains (D1-D5), a transmembrane domain with a single helix, a juxtamembrane domain (JM), and two tyrosine kinase domains (TKDs) [55,56]. The FLT3 receptor binds the FLT3 ligand (FL) through interactions with D3 [55]. The FLT3 receptor exists as a monomer until binding of FL, and dimerization of two receptors promotes phosphorylation of the tyrosine kinase, which in turn activates downstream signaling involved in cell proliferation and activation. Phosphorylation is proposed to be controlled by the autoinhibitory effect of the JM [56]. The internal tandem duplications (ITDs) are in-frame duplication or insertion of 3–1236 nucleotides typically situated in the JM [57]. The TKD mutations are mostly caused by point mutations or small deletions and typically situated in the second tyrosine kinase domain (TKD2) resulting in a single amino acid change or deletion [54]. Among the preclinical anti-FLT3 CAR T-cell therapies reviewed here, Chen et al. [31] and Jetani et al. [30] derived their scFv from the anti-FLT3 antibody clone 4G8, reported by Rappold et al. and Hofmann et al. and it binds Domain 4 on FLT3 [58,59]. Li et al. derived their anti-FLT3 scFV from a monoclonal antibody clone EB10, which binds Domain 4 [60]. Maiorova et al. [36] and Wang et al. [34] used the FLT3 ligand instead of a scFv, which binds to Domain 3. Sommer et al. [33] developed nine anti-FLT3 CARs targeting domain 1–5, and the superior constructs, P3A1 and P3E10, bind to Domains 2 and 4 respectively. Karbowski et al. [32] did not detail the binding location of their scFv.

## Data Availability

Not applicable.

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
