# Peer review of "Recent Advances in the Development of Anti-FLT3 CAR T-Cell Therapies for Treatment of AML"

_biomedicines, 2022, doi:10.3390/biomedicines10102441_

Round 1

Reviewer 1 Report

Recent advances in the development of anti-FLT3 CAR T-cell 2 therapies for treatment of AML By Pedersen is very interesting. However, it needs some additional information to make readers interesting.

Comment 1.  Anti-FLT3 CAR T-cell therapies specific for only AML or for common for other malignancies?

Comment 2.   Anti-FLT3 CAR T-cell does cause any side effects in VIVO?

Comment 3. The authors could expand this Anti-FLT3 CAR T-cell mediated therapy to other cancer models will be very interesting.

Minor: All the in vitro and in vivo need to be italics

Author Response

Recent advances in the development of anti-FLT3 CAR T-cell 2 therapies for treatment of AML By Pedersen is very interesting. However, it needs some additional information to make readers interesting.

We are happy to hear that the reviewer finds the manuscript very interesting. We have addressed the specific issues raised in the comments below.

Comment 1.  Anti-FLT3 CAR T-cell therapies specific for only AML or for common for other malignancies?

Yes, FLT3 amplification has been observed in other malignancies. This has been clarified in the revised manuscript in lines 128-132.

Comment 2.   Anti-FLT3 CAR T-cell does cause any side effects in VIVO?

No toxicities have been identified in the in vivo studies. Potential toxicity issues with killing of CD34+ HSPCs have been described in the manuscript in line 283-299. Safety studies by CAR-T testing in monkeys are described in lines 376-380.

Comment 3. The authors could expand this Anti-FLT3 CAR T-cell mediated therapy to other cancer models will be very interesting.

Indeed, it would be interesting to expand the review to other cancer models, but for the sake of providing a comprehensive and concise review, we have limited the scope to AML.

Minor: All the in vitro and in vivo need to be italics

Thanks, we have observed different journal policies on this matter and will leave it to the typesetters of the journal to change this.

Reviewer 2 Report

This paper describes the development of CAR-Ts directed against FLT-3 and summarizes in vitro and in vivo data to date.  Sufficient background information is given. 

Comments:

1)  Since FLT3L is expressed on B cells, is there concern about immune reconstitution after FLT3 CAR T infusion?

2)Do THP1 cells express FLT3 as their demise was not impressive in some of the experiments (LIne 239)

3)Since some of these are directed to mutational FLT3 and some to WT and some to ligand, it would be helpful to get a perspective on which of these might be effective in an AML setting. 

4)More detail on anticipated barrier to clinical use of these constructs could be provided  Is it anticipated count recovery can occur in their presence, or will transplant be required? 

Minor:

There are several places where pleural vs singular usage is incorrect; eg. line 11 is should be "are" ; line 15 need should be needs, etc. 

Author Response

This paper describes the development of CAR-Ts directed against FLT-3 and summarizes in vitro and in vivo data to date. Sufficient background information is given.

We thank the reviewer for this comment, and address the specific issues raised below.

Comments:

  1. Since FLT3L is expressed on B cells, is there concern about immune reconstitution after FLT3 CAR T infusion?

Yes, there are potentials concerns regarding immune reconstitution, both because of potential elimination of HSPCs, but also because of elimination of cells expressing FLT3, which may eliminate important signaling in the FLT3/FLT3L axis important for immune reconstitution. We have added comments regarding this in the manuscript in lines 141-143 and 384-388.

  1. Do THP-1 cells express FLT3 as their demise was not impressive in some of the experiments (Line 239)

Yes, THP-1 cells have been reported by several studies to express FLT3. In the study referred to, they also validate this. We have clarified this in lines 263-265.

  1. Since some of these are directed to mutational FLT3 and some to WT and some to ligand, it would be helpful to get a perspective on which of these might be effective in an AML setting.

The CAR-T cells described in this review are only directed to FLT3 and not to FLT3L. Neither of the approaches uniquely target the mutational FLT3 or WT FLT3 since the mutations are situated in the intracellular part of FLT3. We have clarified this in lines 455-458.

  1. More detail on anticipated barrier to clinical use of these constructs could be provided. Is it anticipated count recovery can occur in their presence, or will transplant be required?

We thank the reviewer for this question. More details on this matter have been added in lines 503-511.

Minor:

There are several places where pleural vs singular usage is incorrect; eg. line 11 is should be “are”; line 15 should be needs, ect.

We thank the reviewer for bringing this to our attention. This has been corrected.

Reviewer 3 Report

In the review article “Recent advances in the development of anti-FLT3 CAR T-cell therapies for treatment of AML”, Pedersen et al. provided an overview and discussion of different approaches to current anti-FLT3 CAR T-cells under development. It covers the major topics in the field, including CAR construction, efficacy, and safety/toxicity. Overall, it is a well-written, clear, and comprehensive summary of the field of anti-FLT3 CAR T-cell, and it will interest the scientific community.

Minor comments

1.       Adding an introductive or/and a conclusive sentence to each paragraph will help the reader get the points.

2.       Format of the tables should be edited to be suitable for publication. Please use short and descriptive column names. Please do not use shading for column names. “scFv” is not an appropriate name for the second column in Table 1, as some of the CARs are derived from human FLT3 ligand. “Antigen recognition domain” or others could be used.

3.       CAR T-cells targeting several other antigens for AML were mentioned in this manuscript. What are the advantages and disadvantages of FLT3-CAR T-cells compared to those CAR-T cells, such as their efficacy, safety, or toxicity? It will elevate the value of this review to the scientific community and make it more attractive to broader readers.   

Author Response

In this review article “Recent advances in the development of anit-FLT3 CAR-T cell therapies for treatment of AML”, Pedersen et al. provided an overview and discussion of different approaches to current anti-FLT3 CAR T-cells under development. It covers the major topics in the field, including CAR construction, efficacy, and safety/toxicity. Overall, it is a well-written, clear and comprehensive summery of the field of anti-FLT3 CAR T-cell, and it will interest the scientific community.

We thank the reviewer for finding our work well-written and interesting to the scientific community. To improve the manuscript, we have addressed the reviewer’s comments below.

Minor comments:

  1. Adding an introductive or/and conclusive sentence to each paragraph will help the reader get the points.

We thank the reviewer for raising this point. We have added more introductive/conclusive sentences to different paragraphs. Lines 102-105, 171-175, 197-201, 300-303, 382-388.

  1. Format of the tables should be edited to be suitable for publication. Please use short and descriptive column names. Please do not use shading for column names. ”scFv” is not a an appropriate name for the second column in Table 1, as some of the CARs are derived from human FLT3 ligand. ”Antigen recognition domain” or others could be used.

We thank the reviewer for these suggestions, which we have implemented. We will work with the editorial office and publishing team to make the table suitable for publication.

  1. CAR T-cells targeting several other antigens for AML were mentioned in this manuscript. What are the advantages and disadvantages of FLT3-CAR T-cells compared to those CAR-T cells, such as their efficacy, safety, or toxicity? It will elevate the value of this review to the scientific community and make more attractive to broader readers.

This is a good point. We have addressed this in the revised manuscript in lines 479-489.